# Rotation has two sides: Evaluating Data Augmentation for Deep One-class Classification

**Guodong Wang**[1,2]**, Yunhong Wang**[2]**, Xiuguo Bao**[3]**, Di Huang**[1,2*]

[1]State Key Laboratory of Software Development Environment, Beihang University, Beijing, China
[2]School of Computer Science and Engineering, Beihang University, Beijing, China
[3]Natl. Comp. Net. Emer. Resp. Tech. Team/Coord. Ctr. of China, Beijing, China
{wanggd,yhwang,dhuang}@buaa.edu.cn, {baoxiuguo}@139.com

## Abstract

One-class classification (OCC) involves predicting whether a new data is normal or anomalous based solely on the data from a single class during training. Various attempts have been made to learn suitable representations for OCC within a self-supervised framework. Notably, discriminative methods that use geometric visual transformations, such as rotation, to generate pseudo-anomaly samples have exhibited impressive detection performance. Although rotation is commonly viewed as a distribution-shifting transformation and is widely used in the literature, the cause of its effectiveness remains a mystery. In this study, we are the first to make a surprising observation: there exists a strong linear relationship (Pearson's Correlation, $r > 0.9$) between the accuracy of rotation prediction and the performance of OCC. This suggests that a classifier that effectively distinguishes different rotations is more likely to excel in OCC, and vice versa. The root cause of this phenomenon can be attributed to the transformation bias in the dataset, where representations learned from transformations already present in the dataset tend to be less effective, making it essential to accurately estimate the transformation distribution before utilizing pretext tasks involving these transformations for reliable self-supervised representation learning. To the end, we propose a novel two-stage method to estimate the transformation distribution within the dataset. In the first stage, we learn general representations through standard contrastive pre-training. In the second stage, we select potentially semantics-preserving samples from the entire augmented dataset, which includes all rotations, by employing density matching with the provided reference distribution. By sorting samples based on semantics-preserving versus shifting transformations, we achieve improved performance on OCC benchmarks.

## 1 Introduction

One-class classification (OCC) involves determining whether a test sample adheres to the same distribution as the training set, with access only to in-domain data. It represents a classic and fundamental challenge essential for ensuring the reliable deployment of machine learning systems across a broad spectrum of real-world applications, particularly in safety-critical domains like manufacturing defect detection (Bergmann et al., 2020; 2019) and medical diagnosis (Schlegl et al., 2017).

Generative models (Sabokrou et al., 2018; Zaheer et al., 2020; Perera et al., 2019) strive to model the training distribution by assigning high probabilities to training data. However, they often assign unexpectedly high likelihoods to samples that deviate significantly from the training distribution. In contrast, discriminative models encode normality by defining the boundary between the training distribution and out-of-distribution (OOD) samples. In cases where real-world outliers are lacking, one typical solution is to generate negative samples by applying geometric transformations, such as rotation, to the training samples. Contrasting with these transformed samples helps with distinguishing in-domain data from real-world outliers, with these transformations considered as distribution-shifting augmentations (Tack et al., 2020; Sohn et al., 2021) to simulate outliers.

---

*Corresponding author.

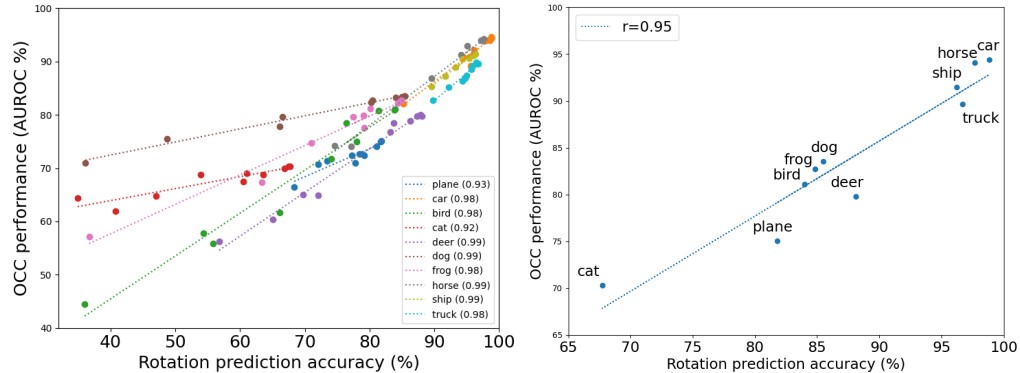

Figure 1: OCC (AUROC) *vs.* accuracy of rotation prediction on one-class CIFAR-10. Figure (a) shows intra-class correlation evaluated every 5 epochs until convergence and (b) displays inter-class correlation for the converged models. Numbers in the bracket denote the Pearson's Correlation $r$). In both intra-class and inter-class experiment, a strong linear correlation ($r > 0.9$) is evident.

Based on this assumption, a plethora of literature has leveraged simulated outliers, albeit with variations in how they are utilized in either pretext tasks or contrastive learning. For instance, (Hendrycks et al., 2019; Golan & El-Yaniv, 2018; Bergman & Hoshen, 2020) train a classifier to identify the specific transformations applied to the original samples. On the other hand, (Sohn et al., 2021; Tack et al., 2020) employ instance discrimination, a special case of contrastive learning, between the original samples and those with distribution-shifting augmentations. This approach is believed to reduce the uniformity of learned representations (Sohn et al., 2021) and promote a more compact inlier distribution (Wang et al., 2023).

For a better understanding of relationship between rotation prediction and OCC, we first set up a toy experiment by training a classifier to predict the degree of rotation applied to input images (ranging from $0°$, $90°$, $180°$, to $270°$). Figure 1 visually represents the connection between the accuracy of rotation prediction and the performance of OCC from both intra-class and inter-class perspectives. All experiments are conducted on the one-class CIFAR-10 benchmark, where images from one class were designated as inliers and those from the remaining classes as outliers. Interestingly, in the intra-class experiment, it is observed that both OCC and rotation prediction accuracies are increasing over the training process, exhibiting a strong linear relationship for each class. This same trend holds true in the inter-class experiment. A similar phenomenon is also noted in unsupervised accuracy estimation (Deng et al., 2021) which utilizes linear regression to estimate classifier performance from the accuracy of rotation prediction. Compared to the work training semantic classification and rotation prediction in a multi-task way, the role of rotation prediction is even prominent in OCC. OCC relies solely on the design of the pretext task to capture normal patterns, with rotation prediction proving to be the most effective. The significant dependence on rotation prediction in OCC motivates us to delve into investigating how rotation impacts OCC.

While rotation has been a widely used technique in the literature for OCC (Wang et al., 2023; Tack et al., 2020; Sohn et al., 2021; Hendrycks et al., 2019; Golan & El-Yaniv, 2018), there has been limited discussion regarding the circumstances under which rotation benefits OCC due to the lack of a detailed analysis on the relationship between rotation and input images. Previous toy experiment (Figure 1) has demonstrated that different classes derive varying benefits from using rotation prediction as a pretext task for OCC. We posit that rotation may not necessarily induce the semantic shift in the original training distribution to the extent previously assumed for different classes. In other words, some transformed samples may still pertain to the same underlying semantic concept. For instance, certain images may lack a deterministic orientation or may deviate from the representative orientations found in the dataset, termed rotation-agnostic images (RAI). Oppositely, images with a deterministic orientation are non-rotation-agnostic images (non-RAI). Based on the division, **rotation generally produces two different results, *i.e.,* rotated RAI are semantics-preserving images while rotated non-RAIs belong to semantics-shifting**.

In the case of RAI, it becomes practically impossible for a classifier to distinguish the original sample from its rotated versions. When such ambiguous images with incorrect labels are incorporated

into training, they inevitably misguide the training process. Similar findings have been reported in prior research (Pal et al., 2020), which introduces the Visual Transformation for Self-Supervision (VTSS) hypothesis. This hypothesis suggests that if the pretext task involves transformations already present in the dataset, the learned representations will be of reduced utility. This naturally raises the question: How can we identify the transformations that are already inherent in the dataset?

Unfortunately, there has been a scarcity of research endeavors aiming at addressing this question. While previous study (Mohseni et al., 2022) can identify a set of shifting transformations from a pool of available options at dataset level, advanced methods provide instance-wise predictions by either leveraging a human-inspected reference subset (Yang et al., 2021) or adopting the early-stop strategy (Miyai et al., 2023). However, both methods depend on human to carefully tune the hyper-parameters, limiting their wide applications.

In this work, we present a novel two-stage framework that addresses the problem from the perspective of distribution matching. In the first stage, we train a model to learn general representations via contrastive pre-training (Chen et al., 2020) and extract representations from all the training images as reference. In the second stage, we learn to sample semantics-preserving images from the entire augmented dataset, which include all transformations, guided by a learnable probability estimator. The goal is to match the distributions of the sampled set and the reference set. During testing, the learned probability estimator assigns high confidence logits to semantics-preserving images and vice versa. This framework allows us to estimate the transformation distribution and effectively distinguish between semantics-preserving and semantics-shifting images.

Our contributions are summarized as follows:

- We establish a connection between rotation prediction and one-class classification, revealing a surprisingly strong linear relationship.
- We introduce a novel two-stage unsupervised framework that estimates the transformation distribution within a dataset and assesses semantic shifting at the instance level.
- By sorting samples based on semantics-preserving versus shifting transformations, we enhance performance on visual one-class classification benchmarks.

## 2 RELATED WORK

**One-class classification (OCC)** entails training a classifier to differentiate anomalous data, which is not accessible during training, from the normal data available in the training set. Unlike anomaly detection (AD), which considers both normal and (optionally) anomalous instances during training, OCC is restricted to only normal instances in its training phase, making it a special case of AD. Generative models (Sabokrou et al., 2018; Zaheer et al., 2020; Perera et al., 2019), trained to model the given distribution, identify test samples located in low-density regions as outliers. However, they often struggle with the curse of dimensionality (Kirichenko et al., 2020). We recommend readers refer to a comprehensive survey on OCC for further details (Perera et al., 2021). In the wake of the success of self-supervised representation learning achieved through methods such as creating pretext tasks (Noroozi & Favaro, 2016; Doersch et al., 2015; Zhang et al., 2016; Gidaris et al., 2018) or employing contrastive learning (He et al., 2020; Chen et al., 2020), discriminative models offer an alternative approach that circumvents the complex process of density estimation and leads to the improved OCC performance. For instance, (Hendrycks et al., 2019; Golan & El-Yaniv, 2018; Bergman & Hoshen, 2020) train a classifier to recognize the transformations applied to the original samples. They rely on this surrogate classifier (*e.g.,* maximum softmax probability as the scoring function) for OCC, under the assumption that outliers cannot be perfectly predicted on rotation angles, as they are not encountered during training. (Tack et al., 2020; Sohn et al., 2021; Wang et al., 2023) have gone further, achieving enhanced results by conducting instance discrimination with advanced anomaly scoring functions. Despite the favorable performance gained with the help of self-labeled generated outliers, these methods manually design discriminative training objectives involving specific transformations.

**Analysis on visual transformations** has primarily focused on the multi-class supervised scenario (Benton et al., 2020; Cubuk et al., 2019; Lim et al., 2019; Mahan et al., 2021; Chatzipantazis et al., 2021) where class boundaries are explicitly defined by manually annotated labels, with limited attention in the challenging one-class classification. The Visual Transformation for Self-Supervision

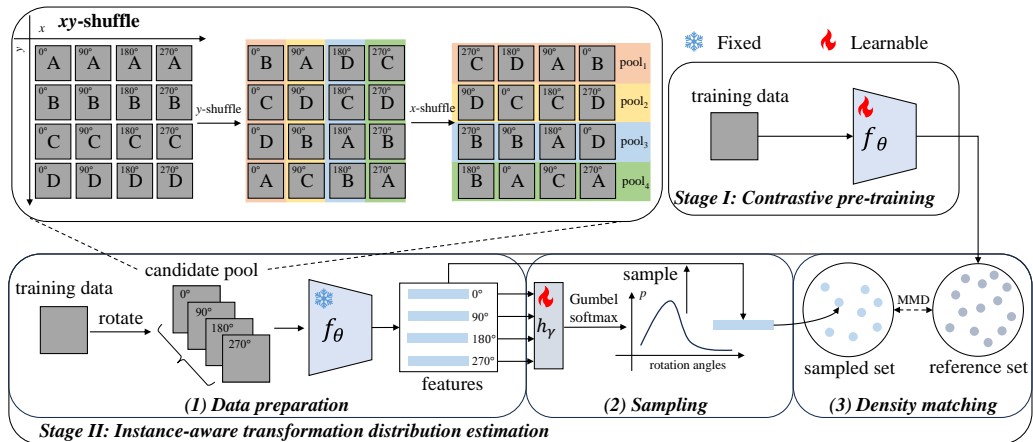

Figure 2: Overview of our two-stage transformation distribution estimation framework. The first stage aims at learning general image representations through contrastive pre-training and the second stage identifies potentially semantics-preserving images by aligning the reference set and the sampled set from from a universal set that encompasses all possible transformations. $xy$-shuffle is proposed for unbiased sampling by reorganizing candidate pools.

(VTSS) (Pal et al., 2020) is the first to raise concerns about the use of potentially conflicting transformations in pretext tasks. In the self-supervised zone, methods addressing transformation conflict are relatively scarce. (Tack et al., 2020) select shifting transformations by measuring the out-of-distribution (OOD) quality, as assessed by area under the receiver operating characteristic curve (AUROC), between in-distribution data and transformed samples using vanilla SimCLR (Chen et al., 2020). Meanwhile, (Mohseni et al., 2022) employ Bayesian optimization (Akiba et al., 2019) to search for effective shifting transformations from a pool of available options through a trial-and-error process. Each trial entails a time-consuming training-validation process, aiming to minimize classification loss on the validation set. However, both of two approaches cannot identify transformation conflicts for each instance within the dataset. A subsequent effort (Yang et al., 2021) introduces instance-wise predictions by utilizing a generative adversarial network (GAN) to automatically discover the transformations present in the input dataset. While this method proves effective for straightforward datasets like MNIST, it heavily relies on a human-inspected reference subset, which is assumed to include the most representative and frequently occurring data to guide the learning of transformation parameter distribution. Consequently, it does not scale well to more complex scenarios due to the unavailability of such a perfect reference subset. An alternative straightforward approach, as proposed by (Miyai et al., 2023; Feng et al., 2019), is to train a rotation predictor before it reaches the point of over-fitting and rely on the classifier to distinguish between RAIs and non-RAIs. However, this early-stop strategy relies on an additional validation set to approximate the epoch just before over-fitting. It necessitates splitting the dataset into multiple consecutive folds and following the standard train/validation split, which can be time-consuming. In contrast, this work focuses on estimating the transformation distribution in an unsupervised manner and determining the likelihood of a transformed sample belonging to the original distribution, addressing these challenges from the perspective of distribution matching.

## 3    TRANSFORMATION DISTRIBUTION ESTIMATION

Existing OCC methods often rely on manually designed transformations to create pretext tasks or perform contrastive learning. These methods assume that these transformations, which are not present in the dataset, suitably shift the original data distribution. Distinguishing between in-domain samples and transformed ones assigns a model with the ability to detect real outliers. However, our empirical observations suggest that the choice of effective data transformations depends on the specific in-domain datasets themselves. Moreover, we find that the learned representations become less effective if the transformations are already present in the dataset, a phenomenon in line with prior work (Pal et al., 2020). To address these issues and discover the transformations that are already

present in the dataset, we propose a staged learning-based framework. This framework enables us to estimate the distribution of transformations and effectively differentiate between semantics-preserving and semantics-shifting images. Without loss of generality, we focus on examining the role of rotation in our main experimental results.

### 3.1 STAGE I: SELF-SUPERVISED REPRESENTATION LEARNING

In line with the recent successes of self-supervised learning for one-class classification (OCC) Tack et al. (2020); Sohn et al. (2021); Wang & Isola (2020), we initially train a model denoted as $g_\phi \circ f_\theta$, which consists of a feature extractor $f_\theta$ and a projection head $g_\phi$, where $\phi$ and $\theta$ denote the corresponding sets of network parameters, using a contrastive pre-training approach (Chen et al., 2020). During the pre-training, positive pairs are defined as two augmented views of the same original image, while negative pairs consist of the augmented versions of all other images. The most-used augmentations to generate positive views for visual representation learning involve random resized cropping, random horizontal flipping, and color jittering, all of which are assumed to be semantics-preserving (Chen et al., 2020; He et al., 2020). Once the model converges, we use the fixed feature extractor $f_\theta$ to extract features from all training images, creating a reference set $\mathcal{D}_{ref} := \{f_\theta(x_i)\}_{i=1}^N$ ($N$ is the dataset size) to supervise the subsequent density matching stage. Notably, different from the existing methods (Tack et al., 2020; Sohn et al., 2021; Wang et al., 2023) that consider rotations as the distribution-shifting transformation and push away the original images from their rotated versions, we exclude rotation from our pre-training process to prevent the rotation itself from becoming a point of examination as a positive or negative augmentation. This strategy ensures distinct representations for images and their rotated counterparts.

### 3.2 STAGE II: TRANSFORMATION DISTRIBUTION ESTIMATION

The second stage comprises three steps: data preparation, sampling, and density matching.

**Data preparation.** The core idea behind estimating the transformation distribution lies in matching density between the original training set and the set of transformed images encompassing all rotation angles. Following the common practice of rotation prediction (Hendrycks et al., 2019; Tack et al., 2020), we consider a four-way classification of rotations, namely $\{0°, 90°, 180°, 270°\}$, and it is easily extended to more classes or a regression task. Formally, given a batch of $M$ images $\mathcal{C} := \{x_i\}_{i=1}^M$, for each $x_i$, we rotate it by four angles to create a set of rotated images that covers all rotations, denoted as $\mathcal{D} := \{R_r(x)|x \in \mathcal{C} \land r \in \{0°, 90°, 180°, 270°\}\}$, where $R_r$ represents a rotation transformation with angle $r$.

**Sampling.** The next question is how to accurately identify images within $\mathcal{D}$ that preserve semantics. On top of the feature extractor $f_\theta$, we introduce a learnable linear layer $h_\gamma$ to estimate the confidence score $s = h_\gamma \circ f_\theta(x)$, $s(x)$ for short, guiding the sampling process. A higher confidence score indicates a higher likelihood of a sample belonging to the input distribution. We first partition $\mathcal{D}$ into a set of $\mathcal{P} := \{p_i\}_{i=1}^M$, where each $p_i := \{R_{0°}(x_i), R_{90°}(x_i), R_{180°}(x_i), R_{270°}(x_i)\}$ acts as a candidate pool and contains four rotated images with different rotation angles. We use a "competing-to-win" strategy to select potentially semantics-preserving images from each $p_i$. Specifically, we employ a differentiable sampler that selects images based on their confidence scores $s$, using a reparameterization technique known as Gumbel softmax (Jang et al., 2016). Note that we choose only one sample from each $p_i$, resulting in a total of $M$ samples constituting the sampled test set $\mathcal{D}_{sp}$ that will be matched with the reference set $\mathcal{D}_{ref}$. Since both stages use the same set of training images, there is a risk that the original sample, *i.e.*, $x_i$ rotated by $0°$, always receives the highest confidence score. To avoid this trivial solution, we introduce the $xy$-shuffle strategy to ensure that each pool $p_i$ contains different images with varying rotation angles and that the sampling is position-irrelevant (Figure 2). Specifically, based on $\mathcal{D}_{sp}$, we first shuffle the images within the same rotation angles and then shuffle the images across different angles. Standard augmentations are also applied to generate diverse positive views, similar to those used in the first stage.

**Density matching.** Density matching is performed in the representation space extracted by the pre-trained model $f_\theta$ in the first stage. For training $h_\gamma$, we utilize an unbiased approximation to the maximum mean discrepancy (MMD) (Gretton et al., 2012) with multiple Gaussian kernels to measure the distribution discrepancy between the sampled set $\mathcal{D}_{sp}$ and the reference set $\mathcal{D}_{ref}$ in the $\ell_2$-normalized representation space. The MMD loss is formally defined as Equation (1).

$$\text{MMD}^2(D_{sp}, D_{ref}) = \frac{1}{n^2} \sum_{i=1}^{n} \sum_{i'=1}^{n} k(x_i, x_{i'}) + \frac{1}{m^2} \sum_{j=1}^{m} \sum_{j'=1}^{m} k(y_j, y_{j'})$$

$$= -\frac{1}{nm} \sum_{i=1}^{n} \sum_{j=1}^{m} k(x_i, x_j) - \frac{1}{nm} \sum_{j=1}^{m} \sum_{i=1}^{n} k(x_j, x_i), \tag{1}$$

where $k(x, y) = \langle \phi(x), \phi(y) \rangle$ is a kernel function and $\phi(\cdot)$ denotes the explicit mapping function. Through the matching of the two distributions, the linear layer $h_\gamma$ tends to assign large logits to semantics-preserving samples.

At inference, by normalizing the confidence scores $s(R_r(x))$ among four rotations of sample $x$, one can easily infer the probability $p(R_r(x))$ of $R_r(x)$ belonging to the input distribution. We use the ensemble strategy with the same augmentations as in training for stable predictions.

$$p(R_r(x)) = \frac{e^{s(R_r(x))}}{\sum_{r \in \{0°, 90°, 180°, 270°\}} e^{s(R_r(x))}}. \tag{2}$$

**Relation to Positive-unlabeled learning (PU-learning).** PU-learning (Mordelet & Vert, 2014; Bekker & Davis, 2020) is a binary classification paradigm where the training data comprises positive examples and unlabeled examples. Our task of identifying transformations present in the dataset can indeed be seen as a specialized instance of PU-learning. In our context, the original set can be considered as positive class, while the transformed images belong to unlabeled set. However, we diverge from conventional PU-learning approaches, such as estimating class priors (Garg et al., 2021) or employing a two-step bootstrap method (Mordelet & Vert, 2014). Instead, we adopt a two-stage pipeline to identify semantics-preserving images from the perspective of distribution matching.

## 4 EXPERIMENTS

### 4.1 EXPERIMENTAL DETAILS

We employ ResNet-18 (He et al., 2016) as the feature extractor $f_\theta$ for both stages, and we operate on the representations right after the global averaging layer. We follow the same augmentation configurations as (Tack et al., 2020) to generate positive views. We train the learnable linear layer $h_\gamma$ for 50 epochs and use stochastic gradient descent (SGD) with momentum as the optimizer and set the learning rate to 0.1. To mitigate biases in the selection of samples, we exclude images from the sampled set when creating the reference subset at each iteration. Our batch size is set to 256, and the temperature $\tau$ in the Gumbel softmax is fixed at 1.0.

### 4.2 VISUALIZATION

We count the number of $\arg\max_{r \in \{0°, 90°, 180°, 270°\}} p(R_r(x))$ for each class and present them as stacked histograms in Figure 3. It is evident that 0° rotation has the highest frequency for all classes in CIFAR-10, indicating a predominant right-side-up orientation among most images. This supports the effectiveness of rotation as a distribution-shifting transformation to generate virtual outliers for OCC (Tack et al., 2020; Sohn et al., 2021; Wang et al., 2023). However, there are exceptions where rotation does not alter the semantics, leading to the uncertain predictions for different rotations, as these images lack a deterministic orientation. With more RAI images involved in training, we observe a linear decrease in OCC performance in Figure 3c. Figure 4 visually demonstrates that our method accurately identifies RAIs and non-RAIs across different classes in both the training and testing sets. It illustrates our method's ability to avoid over-fitting on the training set (Miyai et al., 2023) and to generalize well with unseen test samples. Notably, non-RAIs generally feature centered objects and a zoomed-in appearance, while RAIs capture objects from a distance or in peculiar postures. For instance, RAIs in the plane and bird are primarily captured in low-angle shots with the sky as the background, and rotating these images does not alter their semantics. Conversely, non-RAIs, originally oriented right-side-up on the ground, exhibit unusual postures when rotated, with the ground appearing upside down in the sky.

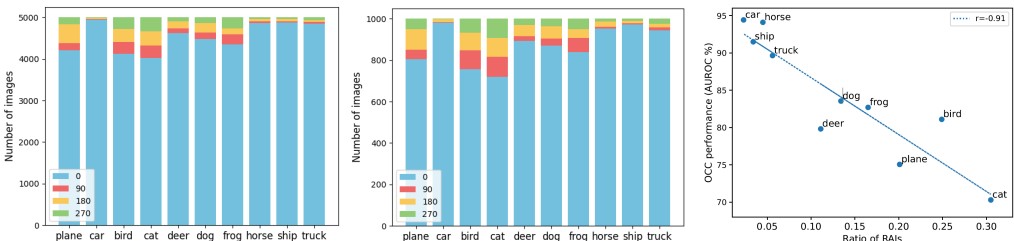

(a) Rot. Distribution in training set. (b) Rot. Distribution in testing set. (c) RAI ratios *vs.* the OCC results.

Figure 3: Stacked histograms for rotation distribution in the training set (a) and testing set (b). The relationship between RAI ratios in the testing set and OCC performance is shown in (c).

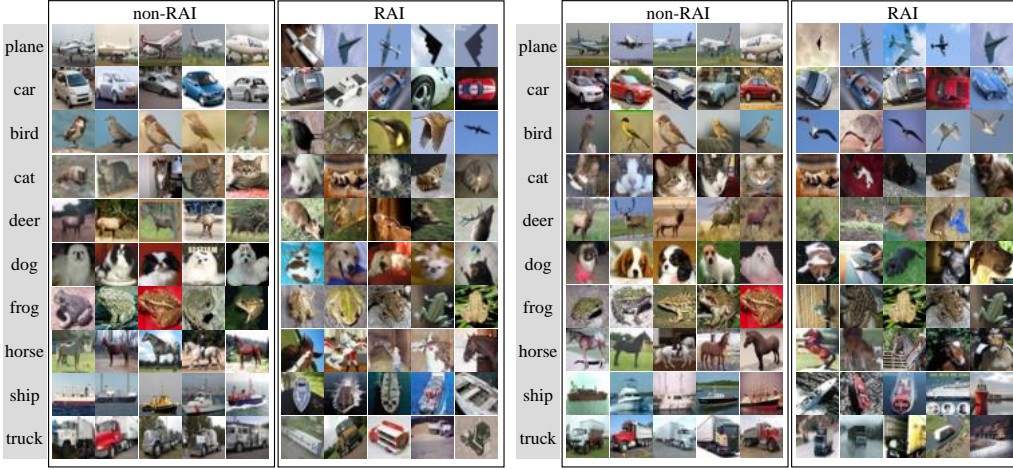

Figure 4: Examples of RAI and non-RAI predicted by our method. Images on the left and right part are from the training and testing sets from CIFAR-10, respectively.

Furthermore, we are interested in how a rotation classifier performs on RAIs/non-RAIs. Intuitively, a well-trained classifier can distinguish between different rotations of non-RAIs but struggles with those of RAIs. For evaluation, we first train a 4-way rotation classifier until convergence and report its accuracy on the testing set in CIFAR-10. Figure 5 displays the rotation accuracies and OCC performance on RAIs and non-RAIs across various classes. This figure highlights two key observations: 1) the classifier achieves significantly lower accuracy on RAIs compared to non-RAIs in nearly all CIFAR-10 classes, indicating accurate classification of RAIs and non-RAIs by our framework; 2) the performance bottleneck in OCC lies in the incorrect rotation prediction for RAIs. Therefore, relying solely on the classification accuracy of the classifier (Hendrycks et al., 2019) for OCC cannot reliably distinguish RAIs from outliers, as both are incorrectly predicted. In such cases, using these images with the misleading labels of rotation angles will misguide the training process.

## 4.3 OTHER TRANSFORMATIONS AND PRE-TRAINING METHODS

While our previous analysis primarily focuses on examining the rotation transformation, it is worth noting that our method is not restricted to rotation alone; it can be applied to various other types of transformations such as vertical/horizontal flip, translation, and composited transformations. Furthermore, our method allows for the utilization of supervised feature extractors, not limited to self-supervised ones. In this section, we evaluate these transformations on the Street View House Numbers (SVHN) dataset (Netzer et al., 2011), where the central digit (0 through 9) in each image is the label. This evaluation provides a better understanding of how transformations shift or preserve semantics, given the clear definition of numbers. We first train the ResNet-18 from scratch in a supervised way on the training set in SVHN, excluding horizontal flip as it is examined in the following stage of distribution estimation. Figure 6 visualizes the estimated distribution of the test set for four

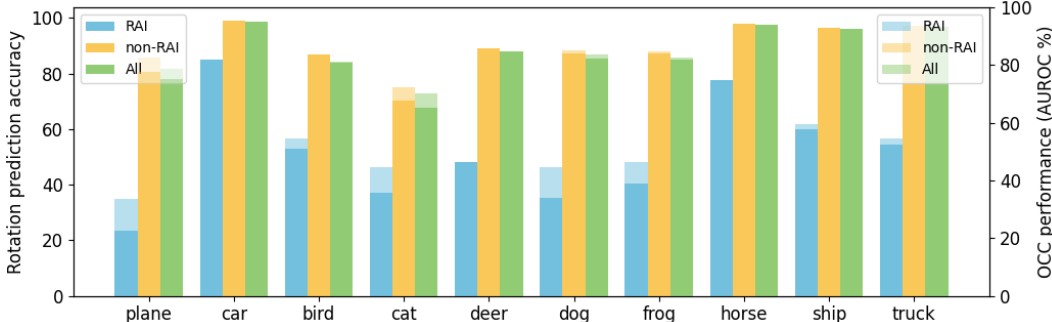

Figure 5: We individually calculate the accuracy of rotation prediction and the OCC performance for RAIs and non-RAIs. Both show a strong link between rotation prediction and OCC.

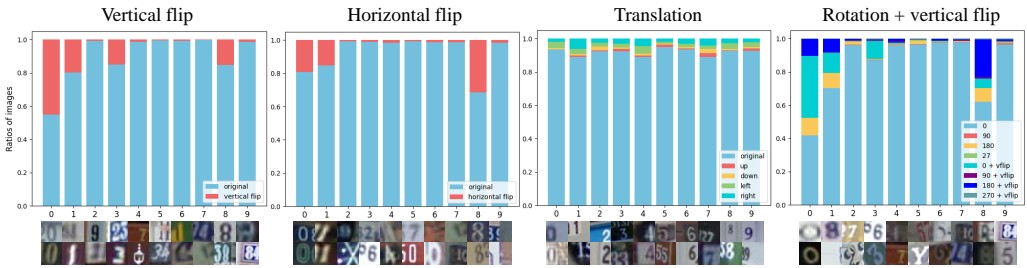

Figure 6: Our method is also applicable to other transformations. Below each sub-figure, the original images in the test set are displayed, with x-axis tick values corresponding to the ground truth specified by the original dataset.

transformations, with stacked histograms below displaying images on which the model makes incorrect predictions. It is evident that different classes exhibit distinct transformation distributions. For flip transformation, nearly half of the images are predicted as vertically flipped for number 0, indicating the involvement of vertical flip, which aligns with common sense. Similarly, numbers 1, 3, and 8 involve vertical flip to some extent, while number 3 lacks invariance to horizontal flip. For translation, we consider five directions (original, up, down, left, right) with an 8-pixel shift, demonstrating the model's ability to correctly identify off-centered digits. Our method is also applicable to composited transformations, where images are rotated followed by a vertical flip. Intriguingly, our method can even spot semantic-shifting images induced by incorrect labeling!

## 4.4 ONE-CLASS CLASSIFICATION

Following (Golan & El-Yaniv, 2018; Sohn et al., 2021), we conduct evaluations on popular OCC benchmarks, including CIFAR-10 (Krizhevsky et al., 2009), CIFAR-100 (20 super-classes) (Krizhevsky et al., 2009), and Cat-vs-Dog (Elson et al., 2007). In these benchmarks, images from one class are treated as inliers, while those from the remaining classes are considered outliers. We utilize AUROC as a threshold-free metric for OCC evaluation.

The preceding analysis highlights that relying solely on a classifier for rotation prediction (Gidaris et al., 2018) is sub-optimal for OCC due to the presence of noisy labels. One straightforward approach is to discard these noisy samples; however, this strategy fails to take full advantage of all the training data and does not capture the comprehensive training distribution. We suggest incorporating additional pretext tasks for representation learning in a multi-task manner beyond rotation prediction. For instance, combining contrastive learning with rotation significantly enhances OCC performance (Tack et al., 2020; Wang et al., 2023). Here, we employ the two one-class classifiers. Before introducing proper modifications, we establish a simple criterion for determining whether a transformed sample is semantically shifted or not, based on the predicted score $p(R_r(x))$. Since the original sample, *i.e.*, $x_i$ without any rotations, serves as trustworthy in-domain data, we take all transformed samples satisfying $p(R_r(x_i)) \geq p(x_i)$ as in-domain (semantics-preserving) data.

(a) One-class CIFAR-10.

| Method | Plane | Car | Bird | Cat | Deer | Dog | Frog | Horse | Ship | Truck | Mean |
|---|---|---|---|---|---|---|---|---|---|---|---|
| Geom (Golan & El-Yaniv, 2018) | 74.7 | 95.7 | 78.1 | 72.4 | 87.8 | 87.8 | 83.4 | 95.5 | 93.3 | 91.3 | 86.0 |
| Rot$^\ddagger$ (Hendrycks et al., 2019) | $78.3_{\pm0.2}$ | $94.3_{\pm0.3}$ | $86.2_{\pm0.4}$ | $80.8_{\pm0.6}$ | $89.4_{\pm0.5}$ | $89.0_{\pm0.4}$ | $88.9_{\pm0.4}$ | $95.1_{\pm0.2}$ | $92.3_{\pm0.3}$ | $89.7_{\pm0.3}$ | 88.4 |
| Rot+Trans$^\ddagger$ (Hendrycks et al., 2019) | $80.4_{\pm0.3}$ | $96.4_{\pm0.2}$ | $85.9_{\pm0.3}$ | $81.1_{\pm0.5}$ | $91.3_{\pm0.3}$ | $89.6_{\pm0.3}$ | $89.9_{\pm0.3}$ | $95.9_{\pm0.1}$ | $95.0_{\pm0.1}$ | $92.6_{\pm0.2}$ | 89.8 |
| GOAD$^\ddagger$ (Bergman & Hoshen, 2020) | $75.5_{\pm0.3}$ | $94.1_{\pm0.3}$ | $81.8_{\pm0.5}$ | $72.0_{\pm0.3}$ | $83.7_{\pm0.9}$ | $84.4_{\pm0.3}$ | $82.9_{\pm0.8}$ | $93.9_{\pm0.3}$ | $92.9_{\pm0.3}$ | $89.5_{\pm0.2}$ | 85.1 |
| iDECODe (Kaur et al., 2022) | $86.5_{\pm0.0}$ | $98.1_{\pm0.0}$ | $86.0_{\pm0.5}$ | $82.6_{\pm0.1}$ | $90.9_{\pm0.1}$ | $89.2_{\pm0.1}$ | $88.2_{\pm0.4}$ | $97.8_{\pm0.1}$ | $97.2_{\pm0.0}$ | $95.5_{\pm0.1}$ | 91.2 |
| CSI (Tack et al., 2020) | $89.9_{\pm0.1}$ | $99.1_{\pm0.0}$ | $93.1_{\pm0.2}$ | $86.4_{\pm0.2}$ | $93.9_{\pm0.1}$ | $93.2_{\pm0.2}$ | $95.1_{\pm0.1}$ | $98.7_{\pm0.0}$ | $97.9_{\pm0.0}$ | $95.5_{\pm0.1}$ | 94.3 |
| CSI + ours | $90.6_{\pm0.1}$ | $99.1_{\pm0.0}$ | $94.4_{\pm0.2}$ | $87.4_{\pm0.1}$ | $94.5_{\pm0.1}$ | $93.5_{\pm0.2}$ | $95.6_{\pm0.0}$ | $98.9_{\pm0.0}$ | $98.1_{\pm0.1}$ | $96.1_{\pm0.2}$ | **94.8** |
| UniCon$^*$ (Wang et al., 2023) | $89.4_{\pm0.2}$ | $99.2_{\pm0.1}$ | $93.3_{\pm0.1}$ | $89.2_{\pm0.0}$ | $94.1_{\pm0.2}$ | $94.2_{\pm0.3}$ | $96.4_{\pm0.0}$ | $98.6_{\pm0.0}$ | $97.7_{\pm0.1}$ | $96.3_{\pm0.2}$ | 94.8 |
| UniCon + ours | $90.0_{\pm0.3}$ | $99.1_{\pm0.0}$ | $93.9_{\pm0.1}$ | $90.4_{\pm0.2}$ | $95.1_{\pm0.0}$ | $95.8_{\pm0.1}$ | $96.8_{\pm0.1}$ | $98.7_{\pm0.0}$ | $97.6_{\pm0.1}$ | $96.8_{\pm0.0}$ | **95.4** |

(b) CIFAR-100 (20 super-classes).

| Method | AUROC |
|---|---|
| GEOM (Golan & El-Yaniv, 2018) | 78.7 |
| Rot (Hendrycks et al., 2019) | 79.7 |
| Rot+Trans (Hendrycks et al., 2019) | 79.8 |
| GOAD (Bergman & Hoshen, 2020) | 74.5 |
| CSI (Tack et al., 2020) | 89.6 |
| CSI + ours | **90.5** |
| UniCon$^*$ (Wang et al., 2023) | 92.0 |
| UniCon + ours | **92.6** |

(c) Cat-vs-Dog.

| Method | Cat | Dog | Mean |
|---|---|---|---|
| RotNet (Golan & El-Yaniv, 2018) | 86.1 | 86.6 | 86.4 |
| Denoising$^\dagger$ Sohn et al. (2021) | 41.3 | 60.6 | 51.0 |
| Contrastive$^\dagger$ Sohn et al. (2021) | 89.7 | 85.7 | 87.7 |
| DROC$^\dagger$ (Sohn et al., 2021) | 91.7 | 87.5 | 89.6 |
| CSI$^*$ (Tack et al., 2020) | 88.9 | 88.6 | 88.8 |
| CSI + ours | **89.5** | **89.3** | **89.4** |
| UniCon$^*$ (Wang et al., 2023) | 92.1 | 89.2 | 90.7 |
| UniCon + ours | **92.5** | **90.0** | **91.3** |

Table 1: AUROC scores on one-class (a) CIFAR-10, (b) CIFAR-100 (20 super-classes) and (c) Cat-vs-Dog. For CIFAR-10, we report the means and standard deviations of AUROC averaged over three trials. $\dagger$ and $\ddagger$ denote the values from DROC (Sohn et al., 2021) and CSI (Tack et al., 2020), respectively. * denotes the reproduced results by ours.

Powerful anomaly detectors like CSI (Tack et al., 2020) and UniCon (Wang et al., 2023) utilize rotation transformations to generate pseudo outliers and learn representations in a contrastive learning framework. However, our analysis demonstrates that rotation does not necessarily alter semantics. Incorporating these noisy images (semantics-preserving but considered outliers) into training disrupts model training, leading to sub-optimal OCC results. By identifying semantic-preserving/shifting images, we modify the training objectives of CSI (Tack et al., 2020) and Uni-Con (Wang et al., 2023) to handle the two types of images separately. For CSI, instead of predicting the label out of four rotation angles for each image, we abstain from making predictions for semantic-preserving images after rotation and only pull close their distances at feature level. For semantic-shifting images, we still predict their rotation labels and promote their separation from other instances, as done in CSI (Tack et al., 2020). For UniCon, we utilize the basic version (Wang et al., 2023) without soft aggregation and hierarchical augmentation tricks for a fair comparison. Rather than promoting the concentration of all original images and dispersion of their rotated versions, we aggregate all semantic-preserving images and push away semantic-shifting ones. We apply the same criterion as (Tack et al., 2020) and (Wang et al., 2023) to compute anomaly scores. Table 1 presents the results on three benchmarks. Building on powerful one-class classifiers, our approach consistently enhances performance across nearly all classes. Notably, the improvement is particularly significant when a class contains more semantic-preserving images w.r.t rotation.

## 5 CONCLUSION

In this study, we initially establish a connection between the widely used rotation prediction task and one-class classification, exhibiting a strong linear relationship. We emphasize the significance of identifying the transformations present in the original dataset before incorporating them into the pretext task for learning. Subsequently, we introduce a novel two-stage framework designed to automatically estimate the distribution of transformations in an unsupervised manner. Leveraging the estimated distribution, we segregate the training data into two distinct sets: one comprising samples without semantics shifting and the other with such shifting. We then address these sets separately for visual representation learning. We believe that our work sheds light on the critical role of understanding the transformations present in the dataset for improving OCC.

**Acknowledgments.** This work is partly supported by the National Key R&D Program of China (2022ZD0161902), the National Natural Science Foundation of China (U20B2069 and 62022011), the Research Program of State Key Laboratory of Software Development Environment (SKLSDE-2023ZX-14), and the Fundamental Research Funds for the Central Universities.

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

**Supplementary Material**

Here, we additionally provide:

- Class-wise performance of RAI and non-RAI binary classification and one-class classification, given manually annotated labels.

- Ablation studies on pre-training dataset, pre-training paradigm and network architecture.

- Evaluation on the multi-class Tiny-ImageNet Le & Yang (2015).

- Evaluation on the real industrial MvTec-AD dataset (Bergmann et al., 2019).

## A  CLASS-WISE PERFORMANCE

### A.1  ACCURACY OF RAI IDENTIFICATION

For quantitative evaluation, binary labels of RAI and non-RAI for the CIFAR-10 training set are manually annotated. The class-wise prediction recall and precision of RAI are reported in the Table A, showing that our method accurately identifies RAIs.

Table A: Class-wise recall and precision on the RAI identification.

| Class | Plane | Car | Bird | Cat | Deer | Dog | Frog | Horse | Ship | Truck | Mean |
|---|---|---|---|---|---|---|---|---|---|---|---|
| RAI ratios (%) | 15.3 | 0.9 | 17.0 | 18.5 | 7.8 | 8.5 | 14.9 | 2.5 | 2.5 | 2.9 | 9.1 |
| Recall | 91.1 | 97.7 | 92.6 | 89.2 | 81.3 | 88.3 | 77.8 | 93.6 | 89.6 | 87.8 | 84.7 |
| Precision | 88.3 | 89.6 | 89.8 | 84.3 | 84.8 | 71.4 | 88.6 | 80.7 | 88.2 | 81.1 | 88.9 |

### A.2  ACCURACY OF OCC ON RAI/NON-RAI

Based on CSI Tack et al. (2020), we present separate results for RAI and non-RAI on one-class CIFAR-10. Table B displays the results for the classes which include a relatively high proportion of RAI samples. Our method significantly enhances OCC performance on RAI, especially for classes with a larger proportion of RAI samples, such as Plane and Cat class.

Table B: Class-wise AUROC scores (%) on OCC performance.

| | Class | Plane | Bird | Cat | Deer | Dog | Frog | Mean |
|---|---|---|---|---|---|---|---|---|
| | RAI ratios (%) | 20.2 | 18.6 | 24.4 | 8.1 | 9.7 | 12.9 | 14.74 |
| CSI | RAI | 73.1 | 90.2 | 79.4 | 92.0 | 91.4 | 92.4 | 86.4 |
| CSI + Ours | RAI | 84.2(+11.1) | 93.1(+2.9) | 85.3(+5.9) | 92.8(+0.8) | 92.5(+1.1) | 92.8(+0.4) | 90.1(+3.7) |
| CSI | non-RAI | 93.2 | 96.5 | 90.8 | 95.2 | 94.9 | 97.4 | 94.7 |
| CSI + Ours | non-RAI | 93.4(+0.2) | 96.5(+0.0) | 91.9(+1.1) | 95.6(+0.4) | 95.2(+0.3) | 97.7(+0.3) | 95.1(+0.4) |

## B  ABLATION STUDY

Our ablation study considers variations in pre-training datasets (CIFAR-10 *vs.* ImageNet-1K), pre-training paradigms (self-supervised *vs.* supervised) and architecture types (ViT-B/16 (Dosovitskiy et al., 2021) *vs.* ResNet-18) in the first stage. Table C demonstrates the architecture-agnostic nature of our method, with ResNet-18 leading to a larger improvement than ViT-B/16. This is attributed to the lack of inductive bias in ViT, which struggles to learn local relations well with a small amount of data, such as CIFAR-10. The results also highlight our method's preference for the supervised training strategy, emphasizing the significance of well-clustered image representations in contrast to uniform distribution (Wang & Isola, 2020) induced by self-supervised training. When pre-training the model on the large-scale dataset, an improved performance is further observed.

Table C: Ablation study on pre-training dataset/paradigm and architecture in the first stage.

| Method | Pre-training dataset | Pre-training paradigm | Architecture | Recall | Precision |
|--------|---------------------|----------------------|--------------|--------|-----------|
| PNDA | CIFAR-10 | Supervised | ResNet-18 | 77.1 | 62.0 |
| Ours | One-class CIFAR-10 | Self-supervised | ViT-B/16 | 82.6 | 83.6 |
| | CIFAR-10 | Supervised | ViT-B/16 | 90.2 | 85.4 |
| | One-class CIFAR-10 | Self-supervised | ResNet-18 | 84.7 | 88.9 |
| | CIFAR-10 | Supervised | ResNet-18 | 93.6 | 90.3 |
| | ImageNet-1K | Supervised | ResNet-18 | **95.5** | **93.5** |

## C MULTI-CLASS CLASSIFICATION

We additionally conduct experiments on the Tiny-ImageNet Le & Yang (2015) which has 100,000 images of 200 classes. As the two-stage pipeline, the first stage pre-trains the encoder by using SimCLR and the second stage learns rotation distribution. Likewise, we use the prediction (Equation (2)) to identify RAI and non-RAI. Given the partition of RAI and non-RAI, we again pre-train the model by using SimCLR, following the practice (Miyai et al., 2023) to deal with RAIs and non-RAIs, *i.e.*, RAIs and their rotations are positive while non-RAIs and their rotations are negative. For fair comparison, we keep the same hyper-parameters in the pre-training. We adopt linear probing and report the top-1 classification accuracy in Table D. We improve the baseline by +2.59%, which is attributed to the enhanced distinction between RAI and non-RAI.

Table D: Linear evaluation on Tiny-ImageNet.

| | RAI ratio (%) | Top-1 accuracy (%) |
|--------|---------------|---------------------|
| PNDA (Miyai et al., 2023) | 31.3 | 37.17 |
| Ours | 28.9 | **39.76** |

## D REAL INDUSTRIAL DATASET

Our experimental evaluation on the MvTec-AD dataset (Bergmann et al., 2019), adopting patch representations of 32x32 built upon UniCon-HA (Wang et al., 2023), demonstrates our method's superiority in Table E compared to counterparts that incorporate the same rotation augmentation. Though our method lags behind the state-of-the-art anomaly detector, we underline the need for tailored pretext tasks (Li et al., 2021) or leveraging pre-trained models (Bergmann et al., 2020) in the context of industrial anomaly detection.

Table E: Results on the MvTec-AD dataset (Bergmann et al., 2019).

| Method | Image | Pixel |
|--------|-------|-------|
| RotNet (Hendrycks et al., 2019) | 71.0 | 92.6 |
| DROC (Sohn et al., 2021) | 86.5 | 90.2 |
| UniCon-HA (Wang et al., 2023) | 89.8 | 94.3 |
| Cut-Paste (Li et al., 2021) | **95.2** | **96.0** |
| Ours | 90.6 | 95.2 |

