# OpenReview forum: "Rotation Has Two Sides: Evaluating Data Augmentation for Deep One-class Classification"
_ICLR.cc/2024/Conference — ICLR 2024 spotlight_

### Official Review · Reviewer_jwSM · 2023-10-16

**Soundness:** 3 good
**Presentation:** 3 good
**Contribution:** 3 good
**Rating:** 6
**Confidence:** 5

**Summary:**

This work targets improving OCC performance by discriminating the RAIs in the training dataset, which is inspired by a surprising observation: there exists a strong linear relationship between the accuracy of rotation prediction and the performance of OCC. The proposed distribution matching-based method is interesting and proved to be effective.

**Strengths:**

1. The idea is novel and interesting.
2. The proposed method is promising and well-analyzed.
3. The paper writing is good.

**Weaknesses:**

1. In the first stage of the proposed method, there is no reason presented for the choice of contrastive pre-training.
2. There are no large-scale dataset evaluations, such as imagenet.
3. There is no direct quantitative evaluation of the RAI predictions. For instance, can the authors annotate a test set with binary labels of whether or not a sample is RAI?

**Questions:**

1. In Page 1 "In cases where real-world outliers are lacking, one typical solution is to generate negative samples by applying geometric transformations, such as rotation, to the training samples.", why are the augmented samples negative?
2. In Sec.3.1, are other pre-trained models suitable for stage 2? Why?
3. In Sec.3.2, how to ensure there is one sample in p_i belonging to the input domain after xy-shuffling?
4. How to pick the RAI samples according to Eq. (2）？
5. There is no definition of p(R_{r}(x)) before it is been used in Eq.(2). And the end of Eq.(2) should be a period.

---

> ### Author Response · Authors · 2023-11-23
> **Response to Reviewer jwSM**
>
> We sincerely thank Reviewer jwSM for providing insightful suggestions. Below, we address your constructive comments individually.
>
>
> **W1**: The reason for the contrastive pre-training in the first stage.
>
> Contrastive pre-training enables the model to extract non-trivial image representations, allowing us to measure the distribution discrepancy in the representation space instead of low-level raw pixel space. The first stage of contrastive pre-training is crucial to make the entire pipeline unsupervised, although the method is not limited to self-supervised feature extractors. Alternatively, using a more powerful feature extractor pre-trained on large-scale datasets in a supervised manner yields improved recall and precision scores. See **Q2** for experimental details.
>
>
> **W2**: Large-scale dataset evaluation.
>
> We conduct experiments on the TinyImageNet which has 100,000 images of 200 classes. As the two-stage pipeline in our manuscript, the first stage pre-trains the encoder by using SimCLR and the second stage learns rotation distribution. Likewise, we use the prediction (Eq.(2)) to identify RAI and non-RAI. Given the partition of RAI and non-RAI, we again pre-train the model by using SimCLR, following the practice [5] to deal with RAIs and non-RAIs, i.e., RAIs and their rotations are postive while non-RAIs and their rotations are negative. For fair comparison, we keep the same hyper-parameters in the pre-training except for the different separations of RAIs and non-RAIs. We adopt liear probing and report the top-1 classification accuracy. We improve the baseline by +2.59% and we attribute the gain to the enhanced distinction between RAI and non-RAI.
>
> | TinyImageNet | RAI ratio (%) |  top-1 accuracy (%) |
> |--------------|---------------|---------------------|
> |     PNDA     |      31.3     |        37.17        |
> |     Ours     |      28.9     |     39.76(+2.59)    |
>
>
> **W3**: Missing quantitative evaluation of the RAI predictions.
>
>
> For quantitative evaluation, binary labels of RAI and non-RAI for the CIFAR-10 training set are manually annotated. The rotation prediction model [3] is initially employed to identify RAI samples incorrectly predicted at 0 degrees, which are then corrected manually as ground-truth RAIs. The remaining samples are designated as non-RAI. The prediction precision and recall of RAI, organized by class, are reported in the table. The table shows that our method can accurately identify RAIs in the training set.
>
> |Class             | plane |  car  | bird |  cat | deer |  dog | frog | horse | ship  | truck | Mean |
> |------------------|-------|-------|------|------|------|------|------|-------|-------|-------|------|
> |RAI ratios (%)    |  15.3 |  0.9  | 17.0 | 18.5 |  7.8 |  8.5 | 14.9 |  2.5  |   2.5 |  2.9  |  9.1 |
> |   Recall         |  91.1 |  97.7 | 92.6 | 89.2 | 81.3 | 88.3 | 77.8 |  93.6 |  89.6 |  87.8 | 84.7 |
> |   Precision      |  88.3 |  89.6 | 89.8 | 84.3 | 84.8 | 71.4 | 88.6 |  80.7 |  88.2 |  81.1 | 88.9 |

---

> > ### Author Response · Authors · 2023-11-23
> > **Continued Response**
> >
> > **Q1**: Why are the augmented samples negative?
> >
> > The augmented samples are considered negative based on the widely-adopted hypothesis in previous works [1, 2, 3, 4]. While acknowledging this hypothesis, our work emphasizes that it does not always hold, depending on the given training dataset. We propose a novel unsupervised approach to identify RAIs and non-RAIs for effective representation learning.
> >
> > **Q2**: In Sec.3.1, are other pre-trained models suitable for stage 2? Why?
> >
> > The model used in the second stage requires two key attributes: 1) the ability to extract meaningful image representations, and 2) distinct representations for images and their rotated counterparts. Therefore, alternative models pre-trained in either a self-supervised or supervised manner can be considered, as long as they satisfy these conditions. The table below shows the results on one-class CIFAR-10 and suggests that our method is applicable for different pre-training approaches. Besides, we also examine a series of transformations on the SVHN dataset using a supervised model (See the Section 4.3 in the updated manuscript). The table shows the results on the training set of CIFAR-10.
> >
> >
> > |  pre-training dataset  |  pre-training way |  Recall |  Precision |
> > |------------------------|-------------------|---------|------------|
> > |   one-class CIFAR-10   |  self-supervised  |  84.7   |    88.9    |
> > |        CIFAR-10        |     supervised    |  93.6   |    90.3    |
> > |      ImageNet-1k       |     supervised    |  95.5   |    93.5    |
> >
> >
> >
> > **Q3**: In Sec.3.2, how to ensure there is one sample in $p_i$ belonging to the input domain after xy-shuffle?
> >
> > We apologize for the confusion. We first perform y-shuffle to ensure each pool $p_i$ has exactly four rotation angles but contains different image instances. Subsequently, x-shuffle is independently performed within each pool. The figure for method has been revised to display xy-shuffle sequentially for clarity.
> >
> >
> > **Q4**: How to pick the RAI samples according to Eq. (2).
> >
> > Images with a prediction of 0 are deemed non-RAI, i.e., $argmax_r{p_r(x))}$ $r \in $ {0,90,180,270} equals 0, while those with any other predictions are considered RAI.
> >
> > **Q5**: There is no definition of p(R_r(x)) before it is been used in Eq.(2). And the end of Eq.(2) should be a period.
> >
> > We have revised them in the manuscript.
> >
> >
> > [1] Tack et al. Csi: Novelty detection via contrastive learning on distributionally shifted instances. In NeurIPS, 2020.
> >
> > [2] Wang et al. Unilaterally Aggregated Contrastive Learning with Hierarchical Augmentation for Anomaly Detection. In ICCV, 2023.
> >
> > [3] Sohn et al. Learning and evaluating representations for deep one-class classification. In ICLR, 2021.
> >
> > [4] Chen et al. Novelty Detection via Contrastive Learning with Negative Data Augmentation. In IJCAI 2021.
> >
> > [5] Miyai et al. Rethinking rotation in self-supervised contrastive learning: Adaptive positive or negative data augmentation. In WACV, 2023.

---

### Official Review · Reviewer_Yjs1 · 2023-10-20

**Soundness:** 2 fair
**Presentation:** 2 fair
**Contribution:** 1 poor
**Rating:** 3
**Confidence:** 4

**Summary:**

Targeting improvement of existing OCC, this paper makes an observation of the strong linear relationship between the rotation prediction and the performance of OCC. To the end, this paper proposes a two-stage framework where in the first stage, standard contrastive learning is used, while in the second stage, semantics-preserving samples are selected from the augmented dataset. Experiments are conducted on several anomaly detection benchmarks.

**Strengths:**

1.	The contribution is clear, and the motivation sounds reasonable.
2.	Analysis on the impact of rotation prediction on OCC is intuitive.
3.	Experiments show the effectiveness of the proposed method.

**Weaknesses:**

1. While the analysis is intriguing, its applicability remains limited to rotation-related datasets and methods. For instance, it may not be suitable for numerous real-world anomaly detection tasks, such as MVTec and VisA.

2. The improvements, as depicted in Table 1, are somewhat modest compared to existing methods.

3. An essential evaluation is missing. This paper identifies RAI images and treats them differently from the original method. However, it is unclear whether the observed improvement stems from these RAI images alone. Assessing the performance of RAI images separately might lead to a more substantial improvement.

4. The paper's structure could be improved. There is significant overlap in the information presented in Figures 1, 2, 4, 5, 6, and 7. It may be advisable to move figures like 4-7 to the supplementary section.

5. The rationale behind the authors' decision to use the version of UniCon without soft aggregation and hierarchical augmentation remains unclear. Since hierarchical augmentation is an integral module within UniCon, it is advisable for the authors to use the full version of UniCon as the baseline for their study.

**Questions:**

See the weakness.

---

> ### Author Response · Authors · 2023-11-23
> **Response to Reviewer Yjs1**
>
> We sincerely thank Reviewer Yjs1 for providing insightful suggestions. Below, we address your constructive comments individually.
>
>
> **W1**: Its applicability remains limited to rotation-related datasets and methods.
>
> Our work extends beyond rotation-related datasets and methods. We additionally conduct experiments on translation distribution prediction in the SVHN dataset (See the Section 4.3 in the updated manuscript). Our experimental evaluation on MvTec-AD, adopting patch representations of 32x32 built upon UniCon-HA, demonstrates our method's superiority compared to counterparts that incorporate rotation augmentation. Though our method lags behind state-of-the-art anomaly detectors, we underline the need for tailored pretext tasks [1] or leveraging pre-trained models [2] in the context of industrial anomaly detection.
>
> The reported Image/pixel-level AUROC scores are as follows:
>
> |         | RotNet | DROC [3] | UniCon-HA [4]| Cutpaste [1] | Ours |
> |---------|--------|----------|--------------|--------------|------|
> |  Image  |   71.0 |   86.5   |      89.8    |     95.2     | 90.6 |
> |  Pixel  |   92.6 |   90.2   |      94.3    |     96.0     | 95.2 |
>
>
>
> **W2**: The improvements (Table 1) are somewhat modest.
>
> Our approach, altering loss functions to handle semantics-preserving/-shifting samples separately, consistently achieves improvements on powerful baselines. While the average improvement over ten classes on one-class CIFAR-10 (Table 1) is not significantly high, it is noteworthy that different classes contain varying RAI ratios. Larger improvements are achieved for classes with a higher proportion of RAIs. For example, for the cat class, which has the largest number of RAIs (see Figure 4(c)), we obtain +1.0% based on CSI and +1.2% based on UniCon.
>
>
> **W3**: Unclear reasons for improvement.
>
> Based on CSI, we present separate results for RAI and non-RAI on one-class CIFAR-10. The table below displays the results for the classes which include a relatively high proportion of RAI samples. Our method significantly enhances OCC performance on RAI, especially for classes with a larger proportion of RAI samples, such as plane and cat.
>
> |              |         |     plane    |    bird    |     cat    |    deer    |     dog    |     frog   |     Mean   |
> |--------------|---------|--------------|------------|------------|------------|------------|------------|------------|
> |RAI ratios (%)|         |  20.2        | 18.6       | 24.4       |  8.1       |  9.7       | 12.9       | 14.74      |
> |   CSI        |  RAI    |  73.1        | 90.2       | 79.4       | 92.0       | 91.4       | 92.4       | 86.4       |
> |   CSI + Ours |  RAI    |  84.2(+11.1) | 93.1(+2.9) | 85.3(+5.9) | 92.8(+0.8) | 92.5(+1.1) | 92.8(+0.4) | 90.1(+3.7) |
> |   CSI        | non-RAI |  93.2        | 96.5       | 90.8       | 95.2       | 94.9       | 97.4       | 94.7       |
> |   CSI + Ours | non-RAI |  93.4(+0.2)  | 96.5(+0.0) | 91.9(+1.1) | 95.6(+0.4) | 95.2(+0.3) | 97.7(+0.3) | 95.1(+0.4) |
>
> **W4**: Paper structure.
>
> Figure 1 and Figure 2 illustrate the correlation between rotation prediction and OCC. Figures 4-7 provide detailed analyses of the effects of RAIs and non-RAIs. We reorganize the structure in our final version.
>
>
> **W5**: Use the complete version of UniCon.
>
> Built upon the complete version of UniCon-HA, which is a very strong OCC detector, it is noted that we also achieve the consistent improvement (+0.2%) on one-class CIFAR-10.
>
>
>
> [1] Li et al. Cutpaste: Self-supervised learning for anomaly detection and localization. In CVPR, 2021.
>
> [2] Paul et al. Uninformed students: Student-teacher anomaly detection with discriminative latent embeddings. In CVPR, 2020.
>
> [3] Sohn et al. Learning and evaluating representations for deep one-class classification. In ICLR, 2021.
>
> [4] Wang et al. Unilaterally aggregated contrastive learning with hierarchical augmentation for anomaly detection, In ICCV, 2023.

---

### Official Review · Reviewer_vKuc · 2023-10-31

**Soundness:** 3 good
**Presentation:** 3 good
**Contribution:** 3 good
**Rating:** 6
**Confidence:** 4

**Summary:**

This paper proposes a technique that can learn the rotation distribution of images within a dataset. The approach is motivated by the authors' study of one-class classification (OCC), which involves predicting whether data belongs to a particular class seen in training or is anomalous. Specifically, the authors are investigating the seeming strong linear relationship between rotation prediction accuracy and OCC. The authors attribute this to transformation bias, where samples that are semantic-preserving vs. semantic-shifting lead to different behavior when training OCC with contrastive learning approaches.

To this end, the authors propose a two-stage approach for learning the transformation distribution: 1) perform a standard contrastive self-supervised representation learning phase, and 2) transformation distribution estimation. For 2, the authors create a dataset consisting of the original samples in {0, 90, 180, and 270} degrees and learn a differentiable sampler with Gumbel Softmax to predict images that preserve semantics (i.e., rotating it does not necessarily change the orientation at which the picture must have been taken). Finally, MMD is used to perform density matching in the representation space to identify such semantic-preserving samples.

The authors use their model to 1) visually show a good model in learning RAIs vs. non-RAIs, 2) a correlation between the amount of RAIs in training and OCC performance, and 3) that using their approach consistently adds ~1% gain to OCC.

**Strengths:**

The strengths of the paper include a clear motivation, strong analysis of the correlation between rotation prediction and OCC, and its unsupervised data-driven approach. The intuition to use a predictor in conjunction with the contrastive pretrained representations and density estimation to align the dataset makes sense for extracting out the transformation distribution exhibited within the training set. Understanding this distribution is an interesting task. The visual results of selecting RAI vs. non-RAI images are compelling. The issues with existing OCC approaches are well-analyzed and the the proposed approach provides modest but consistent improvement to existing OCC approaches.

**Weaknesses:**

One weakness is that there is not a quantifiable way to measure the accuracy in RAI vs. non-RAI determination. One reason for this is that RAI images may be classified as 0, making it hard to separate these images from the truly non-RAI images. There is also not a discussion / inclusion of failure modes to understand where the model may succeed vs. fail. Another weakness is that the utility of the transformation distribution may be larger but is focused on OCC, and datasets used to evaluate OCC are from CIFAR-10, which may have different characteristics than some of the anomaly detection settings where labeled data of out-of-distribution samples could be limited. I also think it is odd that OCC is a common thread / motivator of the paper but the description of the technique for OCC is not in the approach section. The claim of the paper then is a bit broad, in some parts reading as if it is most concerned about OCC and other parts about the distribution being learned.

**Questions:**

1. In Section 4.2: it seems that images with a prediction of 0 is deemed non-RAI and anything else is deemed to be RAI. Couldn't RAI images still be predicted with an angle of 0? How many of the RAI images are classified as angle 0 and how many images classified as angle 0 are actually RAI?

2. Do you have any examples / analysis of failure modes where images like those in figure 5 are mistakenly predicted to be RAI vs. non-RAI?

3. Can you clarify the rule used in section 4.4 to determine if a sample is semantically-shifted? Which r is used?

---

> ### Author Response · Authors · 2023-11-23
> **Response to Reviewer vKuc**
>
> We sincerely thank Reviewer vKuc for providing insightful suggestions. Below, we address your constructive comments individually.
>
> **W1(Q1)**: Quantitative accuracy in RAI vs. non-RAI determination.
>
> For quantitative evaluation, binary labels of RAI and non-RAI for the CIFAR-10 training set are manually annotated. The rotation prediction model [3] is initially employed to identify RAI samples incorrectly predicted at 0 degrees, which are then corrected manually as ground-truth RAIs. The remaining samples are designated as non-RAI. The prediction precision and recall of RAI, organized by class, are reported in the table. The table shows that our method can accurately identify RAIs. The results can be further improved by leveraging a more powerful pre-trained feature extractor, as demonstrated in **W2**.
>
>
> Trainig set:
> |Class             | plane |  car  | bird |  cat | deer |  dog | frog | horse | ship  | truck | Mean |
> |------------------|-------|-------|------|------|------|------|------|-------|-------|-------|------|
> |RAI ratios (%)    |  15.3 |  0.9  | 17.0 | 18.5 |  7.8 |  8.5 | 14.9 |  2.5  |   2.5 |  2.9  |  9.1 |
> |   Recall         |  91.1 |  97.7 | 92.6 | 89.2 | 81.3 | 88.3 | 77.8 |  93.6 |  89.6 |  87.8 | 84.7 |
> |   Precision      |  88.3 |  89.6 | 89.8 | 84.3 | 84.8 | 71.4 | 88.6 |  80.7 |  88.2 |  81.1 | 88.9 |
>
>
>
> **W2(Q2)**: Failure modes.
>
> In fact, the last table shows that our model has misclassified images and these incorrect predictions stem from imperfect representations learned in contrastive pre-training, which are contextually biased towards spurious scene correlations [1, 2]. To address this, we suggest using a more powerful feature extractor pre-trained on large-scale datasets. For instance, leveraging an encoder pre-trained on ImageNet-1k has resulted in improved outcomes in the identification of both RAI and non-RAI. The table shows the results on the training set of CIFAR-10.
>
> |  pre-training dataset  |  pre-training way |  Recall |  Precision |
> |------------------------|-------------------|---------|------------|
> |   one-class CIFAR-10   |  self-supervised  |  84.7   |    88.9    |
> |        CIFAR-10        |     supervised    |  93.6   |    90.3    |
> |      ImageNet-1k       |     supervised    |  95.5   |    93.5    |
>
>
>
>
> **W3**: Transformation distribution vs. OCC.
>
> While rotation prediction as a pretext task has been explored in the self-supervised literature and facilitates bundles of downstream tasks, its significance becomes more pronounced in OCC, where the effectiveness relies solely on the design of the pretext task to capture normal patterns. Furthermore, our empirical experiments reveal a strong linear relationship between the accuracy of rotation prediction and the performance of OCC. This observation motivates us to delve into investigating how the separation between RAI and non-RAI impacts OCC. While we have achieved improved results in multi-class classification on TinyImageNet, evaluated by linear probing. We have revised the manuscript to provide additional clarity on this aspect.
>
> | TinyImageNet | RAI ratio (%) |  top-1 accuracy (%) |
> |--------------|---------------|---------------------|
> |     PNDA     |      31.3     |        37.17        |
> |     Ours     |      28.9     |     39.76(+2.59)    |
>
> **Q1**: Determination of RAI and non-RAI.
> Yes. Images with a prediction of 0 are deemed non-RAI, i.e., $argmax_r{p_r(x))}$ $r \in $ {0,90,180,270} equals 0, while those with any other predictions are considered RAI.
>
> **Q3**: Determination of a semantically-shifting sample. Which $r$ is used?
>
> Semantically-shifting samples are determined by comparing the original image x and its rotated version $R_r(x)$ ($R_r$ denotes rotating $x$ by $r$ degrees, $r$ belongs to {0, 90, 180, 270}). An image $R_r(x)$ satisfying $p(R_r(x)) < p(x)$ is considered semantically-shifting. The determination is conducted by considering all four rotation angles. Note that for a RAI sample, at least one rotated sample is semantically-preserving.
>
>
> [1] Mo et al. Object-aware Contrastive Learning for Debiased Scene Representation. In NeurIPS 2021.
>
> [2] Purushwalkam et al. Demystifying Contrastive Self-Supervised Learning: Invariances, Augmentations and Dataset Biases. In NeurIPS, 2020.
>
> [3] Miyai et al. Rethinking rotation in self-supervised contrastive learning: Adaptive positive or negative data augmentation. In WACV, 2023.

---

### Official Review · Reviewer_Soam · 2023-11-01

**Soundness:** 2 fair
**Presentation:** 3 good
**Contribution:** 3 good
**Rating:** 6
**Confidence:** 3

**Summary:**

The study makes a surprising discovery: a strong linear relationship exists between the accuracy of rotation prediction and the performance of OCC and they show that representations learned from transformations already present tend to be less effective. To address this, the paper proposes a staged learning-based framework for one-class classification (OCC) that aims to identify semantics-preserving images. The framework consists of two stages: self-supervised representation learning and transformation distribution estimation.

**Strengths:**

- I think the authors do a great job at explaining and motivating the problem they are working on.
- The paper seems novel in its exploration of the relationship between rotation prediction and one-class classification (OCC). It highlights a surprising observation of a strong linear relationship between the performance of rotation prediction and the performance of OCC.
- The authors back up their approach on empirical observations, highlighting the importance of effective data transformations and the potential decrease in effectiveness if transformations are already present in the dataset. This empirical foundation strengthens the credibility of their proposed solution. Their approach seems to be very well-motivated
- The experiments though little are well-designed, valid, and exhaustive, with comparison to a range of baselines as well as some ablation studies.

**Weaknesses:**

- A big weakness right now is the lack of extensive empirical validation. The authors currently only perform experiments on CIFAR-10 and their experiments on other kinds of transformations are also very limited. Though the authors show interesting results for another transformation, it is immensely difficult with this set of results to comment on how well their approach could work.
- One of the really interesting findings from this paper is about transformation bias, and representations learned from transformations already present tend to be less effective. I believe this would be well-shown by experiments across multiple datasets and multiple kinds of models.

**Questions:**

- How does the analysis across other kinds of popular transformations look like and does this approach still hold, I would encourage the authors to include talking about other transforms even if they do not seems to work well.
- This shouldn't use in-text citations

> facturing defect detection (Bergmann et al., 2020; 2019) and medical diagnosis Schlegl et al. (2017).

---

> ### Author Response · Authors · 2023-11-23
> **Response to Reviewer Soam**
>
> We sincerely appreciate your recognition of the novelty and motivation behind our work. Below, we address your constructive comments individually.
>
> **W1**: Lack of empirical validation on datasets other than CIFAR-10.
>
> We additionally conduct experiments on the TinyImageNet which has 100,000 images of 200 classes. As the two-stage pipeline in our manuscript, the first stage pre-trains the encoder by using SimCLR and the second stage learns rotation distribution. Likewise, we use the prediction (Eq.(2)) to identify RAI and non-RAI. Given the partition of RAI and non-RAI, we again pre-train the model by using SimCLR, following the practice [5] to deal with RAIs and non-RAIs, i.e., RAIs and their rotations are postive while non-RAIs and their rotations are negative. For fair comparison, we keep the same hyper-parameters in the pre-training except for the different separations of RAIs and non-RAIs. We adopt liear probing and report the top-1 classification accuracy. We improve the baseline by +2.59% and we attribute the gain to the enhanced distinction between RAI and non-RAI.
>
> | TinyImageNet | RAI ratio (%) |  top-1 accuracy (%) |
> |--------------|---------------|---------------------|
> |     PNDA     |      31.3     |        37.17        |
> |     Ours     |      28.9     |     39.76(+2.59)    |
>
> For the experiments on SVHN, refer to responses to W2 for details.
>
>
> **W2 (Q1 and Q2)**: Other transformations.
>
> To address the need for exploring other transformations, we have extended our evaluation to include translation on the Street View House Numbers dataset (SVHN). The goal is to classify the central digit in each image into one of the categories (0 through 9). We consider five translation directions (original, up, down, left, right) with an 8-pixel shift. Notably, our approach is not confined to self-supervised pre-training. The first stage trains the ResNet-18 from scratch in a supervised way, achieving 96.24% accuracy on the test set. The second stage predicts the 5-way translation distribution. As shown in the updated manuscript, it showcases the model’s ability to correctly identify off-centered digits. See Section 4.3 in the updated manuscript for more discussion on the experiments on SVHN.
>
>
> **W3**: Experiments across other models.
>
> We have explored ViT-based models, specifically ViT-B/16, for rotation distribution prediction in CIFAR-10. Additionally, we considered variations in architecture types and pre-training strategies (self-supervised vs. supervised). For quantitative evaluation, binary labels of RAI and non-RAI for the CIFAR-10 training set are manually annotated. The provided table demonstrates the architecture-agnostic nature of our method, with ResNet-18 leading to a larger improvement than ViT-B/16. This is attributed to the lack of inductive bias in ViT, which struggles to learn local relations well with a small amount of data, such as CIFAR-10. The results also highlight our method's preference for the supervised training strategy, emphasizing the significance of well-trained image representations, as indicated by average recall and precision values across the 10 classes in CIFAR-10.
>
>
> | Architecture | pre-train (1st stage) |   Recall  |  Precision  |
> |--------------|-----------------------|-----------|-------------|
> |    PNDA[1]   |           /           |   77.1    |    62.0     |
> |   ResNet-18  |    self-supervised    |   84.7    |    88.9     |
> |   ResNet-18  |      supervised       |   93.6    |    90.3     |
> |   ViT-B/16   |    self-supervised    |   82.6    |    83.6     |
> |   ViT-B/16   |      supervised       |   90.2    |    85.4     |
>
>
>
>
>
> [1] Miyai et al. Rethinking rotation in self-supervised contrastive learning: Adaptive positive or negative data augmentation. In WACV, 2023.

---

### Official Review · Reviewer_h8uB · 2023-11-05

**Soundness:** 3 good
**Presentation:** 2 fair
**Contribution:** 3 good
**Rating:** 6
**Confidence:** 2

**Summary:**

This paper introduces a novel method to improve performance on one-class classification problem. They observe a strong correlation between the accuracy of rotation prediction and one-class classification. They introduce a two-stage unsupervised framework that differentiates rotations that are semantic preserving (rotation-agnostic images) vs semantic shifting (non-rotation agnostic images) to enhance performance on the one-class classification benchmarks.

**Strengths:**

- The method introduced detects rotations that remain unchanged and are semantically similar to original images, ensuring their exclusion as outliers. This method demonstrates sufficient generalizability for other transformation (ex., gaussian noise), as supported by section 4.3. I feel that this contribution is of sufficient interest to the research community.
- The results on OCC presented in the paper outperform the baselines and can be added on top on existing methods. It emphasizes the significance of identifying the transformations present in the original dataset before incorporating them into the pretext task for learning.

**Weaknesses:**

- Related works section of the paper is difficult to follow and seems incomplete.
    - The subheading of **One-Class Classification** is particularly confusing as it lacks comprehensive discussion on the relevant OCC literature. The authors directly diverge to self-supervised learning methods for OCC. It fails to give complete picture of OCC for non-experts. I would suggest the authors to briefly also give an introduction to OCC and existing generative methods or point readers to more detailed survey paper [1].
    - There exists a confusion regarding anomaly detection and OCC cited in related work, in which cases are they both considered the same?
- Missing relevant citation: the authors seem to be missing an important citation on rotation estimation [2]. I would suggest the authors to include it for completeness of related work.

[1] One-Class Classification: A Survey, arxiv 2021

[2] Self-Supervised Representation Learning by Rotation Feature Decoupling, CVPR 2019

**Questions:**

- My major suggestions are summarized in Weakenesses section
- In introduction: “While rotation has been a widely used technique in the literature for OCC…” missing citations, please add them here?

---

> ### Author Response · Authors · 2023-11-23
> **Response to Reviewer h8uB**
>
> Thank you for appreciating and acknowledging our work. We address your constructive comments below:
>
> **W1**: Related works.
>
> We have incorporated more discussion on relevant OCC literature in the revised manuscript. Anomaly detection (AD) and OCC share similarities, aiming to identify instances deviating from normal training data and providing a binary classifier to determine normality or anomaly. The key distinction lies in the training data characteristics. AD considers both normal and anomalous instances during training, while OCC is confined to only normal instances in training. Therefore, OCC can be regarded as a special case of AD. When all training data belongs to one class, AD and OCC methods are expected to yield similar results.
>
>
> **W2(Q2)**: Missing citations.
>
> We appreciate the reviewer's suggestion for additional references. We previously noted that [1] uses the similar idea with [2] to identify RAI and non-RAI by training a binary classifier on the noisy dataset while [2] improves the idea by leveraging an extra validation set to approximate the epoch just before over-fitting. We have included it and discussed its relevance in the revised version.
>
> [1] Fent et al. Self-Supervised Representation Learning by Rotation Feature Decoupling. In CVPR, 2019.
>
> [2] Miyai et al. Rethinking rotation in self-supervised contrastive learning: Adaptive positive or negative data augmentation. In WACV, 2023.

---

### Author Response · Authors · 2023-11-23
**Common response**

We appreciate the detailed and constructive feedback from all the reviewers. Specifically, we are pleased that they acknowledge the work's sufficient (h8uB) and clear (Yjs1) contribution. Additionally, the reviewers find the idea to be novel (Soam, jwSM) and interesting (vKuc, jwSM), supported by a clear motivation (vKuc) and good writing (jwSM). The method is both well-motivated (Soam) and well-analyzed (jwSM). Before addressing the individual reviews, we briefly outline the manuscript changes and recurring points highlighted in the feedback.


1. Enhanced Related Work Discussion:
	- We have expanded the discussion on relevant OCC literature.

2. Quantitative Evaluation:
	- Binary labels of RAI and non-RAI for the CIFAR-10 testing and training sets are manually annotated.
	- Using the ground-truth, we calculate the accuracy of RAI and non-RAI classification and report OCC performance separately for RAI and non-RAI.

3. Extended Evaluation:
	- Datasets: Evaluation is now conducted on three additional datasets: TinyImageNet, MvTec-AD, and SVHN.
	- Network Architectures: ViT networks are additionally considered as encoders.
	- Pre-training Methods: In addition to contrastive pre-training in the first stage, a supervisedly trained model is also considered as an encoder.
	- Transformations: Translation on the SVHN dataset is examined.
	- Multi-class Evaluation: The effectiveness of the distinction between RAI and non-RAI is evaluated, reporting accuracy in multi-class classification on TinyImageNet.

4. Minor Edits:
	- Typos have been corrected, and grammar has been enhanced.

---

### Meta-Review · Area_Chair_cnjY · 2023-12-12

**Metareview:**

This paper discovers that the accuracy of rotation prediction is strongly correlated with one-class classification (OCC), based on which a novel method is proposed for estimating the transformation distribution within the dataset, thereby enhancing OCC.

The strengths of the paper are novel insight, good writing, and consistent performance gains.  The weaknesses of the paper are modest performance gains,  insufficient experimental validation with respect to a variety of datasets and transformations (other than rotation), and whether rotation-agnostic images (RAI) are a significant part in a real dataset for such a method to have a performance impact.

The paper has received 4 reviews, with ratings 6/6/3/6/6, to which authors have provide further clarification and additional experiments on more substantial datasets and other transformations.  The only negative review has a lingering concern on the significance of RAI, while all reviewers consider the basic finding surprising, novel, and interesting.  Based on the extensive rebuttals and reviewers' detailed comments, the AC agrees with the majority consensus and recommends acceptance.

**Justification For Why Not Higher Score:**

Lack of substantial experimental validation.
Limited performance gains.

**Justification For Why Not Lower Score:**

Novel insight that connects natural transformation distributions to recognition performance.
Develop a technical method to identify such transformations.

---

### Decision · Program_Chairs · 2024-01-16

Accept (spotlight)